# A MOTH BRAIN LEARNS TO READ MNIST

**Charles B. Delahunt**
Department of Electrical Engineering
University of Washington
Seattle, WA, 98195, USA
delahunt@uw.edu

**J. Nathan Kutz**
Department of Applied Mathematics
University of Washington
Seattle, WA, 98195, USA
kutz@uw.edu

## ABSTRACT

We seek to characterize the learning tools (ie algorithmic components) used in biological neural networks, in order to port them to the machine learning context. In particular we address the regime of very few training samples.

The Moth Olfactory Network is among the simplest biological neural systems that can learn. We assigned a computational model of the Moth Olfactory Network the task of classifying the MNIST digits. The moth brain successfully learned to read given very few training samples (1 to 20 samples per class). In this few-samples regime the moth brain substantially outperformed standard ML methods such as Nearest-neighbors, SVM, and CNN.

Our experiments elucidate biological mechanisms for fast learning that rely on cascaded networks, competitive inhibition, sparsity, and Hebbian plasticity. These biological algorithmic components represent a novel, alternative toolkit for building neural nets that may offer a valuable complement to standard neural nets.

## 1 INTRODUCTION

Neural net (NN) architectures have achieved strong success in a wide array of machine learning (ML) tasks [Schmidhuber (2015)]. But they are also known to fail on critical tasks such as learning from few samples. We seek to improve NN performance on such tasks by characterizing mechanisms of biological neural networks (BNNs) involved in learning.

The Moth Olfactory Network is among the simplest BNNs that can learn [Riffell et al. (2012)], yet it contains key features widespread in BNNs: High noise [Galizia (2014)], random connections [Caron et al. (2013)], Hebbian synaptic growth [Cassenaer & Laurent (2007)], high-dimensional sparse layers [Campbell & Turner (2010)], large dimension shifts between layers [Ganguli & Sompolinsky (2012)], and generalized stimulation of neurons during learning [Hammer & Menzel (1995)].

[Delahunt et al. (2018)] developed an end-to-end computational model, MothNet, of the *Manduca sexta* moth olfactory network. The model is closely based on known biophysical structure, is consistent with *in vivo* firing rate data, and incorporates learning dynamics. We gave MothNet the classic ML task of identifying the handwritten digits of the MNIST dataset [LeCun & Cortes (2010)]. MothNet routinely achieved 75% to 85% accuracy classifying test digits after training on 1 to 20 samples per class. In this few-samples regime it substantially out-performed standard ML methods such as Nearest-neighbors, SVM, and CNN (Fig 2). The results demonstrate that even very simple biological architectures hold novel and effective algorithmic tools applicable to ML tasks, in particular tasks constrained by few training samples or the need to add new classes without full retraining.

## 2 METHODS

**MothNet model:** The feed-forward moth olfactory system (schematic Fig 1) involves two interacting networks, the noisy antennal lobe (AL) and the sparse mushroom body (MB) [Wilson (2008); Campbell & Turner (2010)]. The AL contains 60 processing units, onto which atomic olfactory features map one-to-one. Competitive inhibition between these units decorrelates input class signatures by sharpening contrasts between input projections onto the AL. Learning occurs when a reward (sugar) induces an overall increase in excitation in the AL via the neuromodulator octopamine

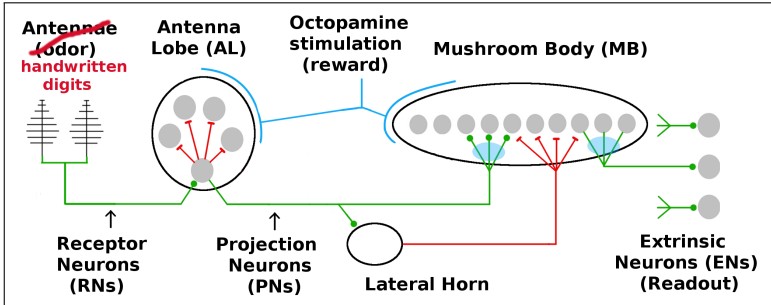

Figure 1: **Network schematic.** Green lines show excitatory connections, red lines show inhibitory connections. Light blue ovals show plastic connections into and out of the MB. The processing units in the AL competitively inhibit each other. Global inhibition from the lateral horn induces sparsity on MB responses. The ENs give the final, actionable readouts of the system's response to a stimulus.

[Dacks et al. (2012)]. This induces Hebbian updates to weights in the plastic MB, i.e. modulating one network rewires another. Sparsity in the MB controls plasticity by filtering noise and thus focuses training gains on relevant signals. In combination, these features enable rapid and effective learning, expressed as modulated readout neuron responses. Details in [Delahunt et al. (2018)].

The MothNet model closely follows the moth's olfactory architecture in terms of connections, numbers of neurons in each layer, etc. Neural firing rates are modeled with integrate-and-fire dynamics [Dayan & Abbott (2005)] evolved as stochastic differential equations (SDEs) [Higham. (2001)]:

$$\tau \frac{dx}{dt} = -x + s(\Sigma \mathbf{w}_i \mathbf{u}_i) = -x + S(\mathbf{w} \cdot \mathbf{u}) + dW, \tag{1}$$

where $x(t)$ = firing rate (FR) for a neuron, $\mathbf{w}$ = connection weights, $\mathbf{u}$ = upstream neuron FRs, $S()$ is a sigmoid function or similar, and $W(t)$ = a brownian motion process. Hebbian plasticity governs synaptic weight updates [Hebb (1949)]:

$$\Delta w_{ab}(t) = \gamma f_a(t) f_b(t) \tag{2}$$

where $f_a(t)$, $f_b(t)$ are firing rates of neurons $a$, $b$ at time $t$; $w_{ab}$ is the synaptic weight between them; and $\gamma$ is a growth rate parameter. Inactive MB→EN weights are subject to proportional decay:

$$\Delta w_{ab}(t) = \delta w_{ab}(t), \text{ if } f_a(t) f_b(t) = 0. \tag{3}$$

where $\delta$ is a decay parameter. There are two layers of plastic synaptic weights: AL→MB, and MB→ENs (ie pre- and post-MB), both controlled by sparsity in the MB.

Ten readout neurons (ENs) are randomly assigned to target particular digits 0 to 9. Training is supervised: When a digit of class $j$ is presented, only MB→EN$_j$ connections are updated, where EN$_j$ is the EN assigned to class $j$. That is, the moth knows the class of the training sample, for the purposes of post-MB updates. Training rapidly tailors these weights to their target digits.

**Training data:** The MNIST dataset consists of grey-scale thumbnails, 28 x 28 pixels, of handwritten digits 0 - 9. We used pixels of the thumbnails as input features to the MothNet classifier. These provide a good test of whether a system can effectively learn to discriminate classes given inputs with high inter-class correlations.

In the moth, input features map one-to-one into 60 AL processing units. To reduce the number of pixels to this scale, we (i) downsampled by 2; (ii) subtracted a population mean; (iii) killed negative values; (iv) then selected only the most-generally-active pixels. This gave thumbnails with 83 retained pixels. Each pixel was a feature that fed into one processing unit of MothNet's AL. Thumbnails for each experimental stage (mean-subtraction, pre-training baseline, training, post-training test set) were randomly chosen without replacement from non-intersecting pools.

**Classifier:** System readout units (ENs) are silent absent any input sample, and they consistently respond, more or less strongly, to input samples. We classified test digits using a summed log-likelihood over the distributions of responses to each digit class in each EN:

$$\hat{s} = \min_{j \in J} \{ \sum_{i \in J} (\frac{E_i(s) - \mu E_{ij}}{\sigma E_{ij}})^4 ) \}, \text{ where} \tag{4}$$

$\hat{s}$ = predicted class of sample $s$; $E_i(s)$ = response of the $i$th EN to $s$; $\mu E_{ij} = \text{mean}(E_i(t)|t \in V, t \in \text{class } j)$; $\sigma E_{ij} = \text{std dev}(E_i(t)|t \in V, t \in \text{class } j)$; $j \in J$ are the classes (0-9); $V$ is a reference set (eg a validation set). Roughly, $j$ is a strong candidate for $\hat{s}$ if each EN's response to $s$ is close to that EN's expected response to class $j$. The use of the $4^{\text{th}}$ power is a sharpener that penalizes outliers.

## 3 RESULTS

**MothNet learning behavior:** Moths randomly generated from MothNet templates responded consistently well to training by differentiating their various EN responses to different digit classes. Training caused EN responses to diverge from baseline and from each other, such that each EN responded most strongly to its assigned digit. "Natural" moths routinely achieved 75% to 85% accuracy given 15 - 20 training samples. "Fast" moths, ie with Hebbian growth rate parameters "turned up to 11", achieved 70% accuracy given just one training sample, but with no further gains (Fig 2).

**Comparison to standard ML methods:** As baseline we used three standard ML methods (Nearest-neighbors, SVM, and CNN), optimized for each number-of-training-samples case. While these ML methods can attain over 99% accuracy on the full MNIST training set (6000 samples per class) [LeCun et al. (1995)], the few-samples regime is fundamentally different, and standard ML methods are not well suited to it, compared to biological systems. This few-samples regime can be roughly visualized, for MNIST, as follows: First throw away 99.9% of the usual training data; then begin training.

In this few-samples regime, MothNet substantially out-performed the three ML methods. Given $\leq 5$ training samples, MothNet had roughly double the accuracy of ML methods. To reach equivalent accuracy, ML methods required between 2x and 10x more training data than the "natural moth", and between 20x to 50x more training data than the "fast moth". Mean trained accuracies of the various methods are plotted in Fig 2, vs number of training samples per class (log scale).

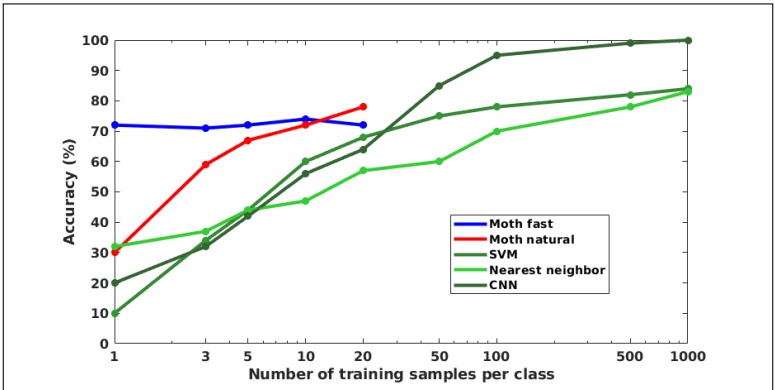

Figure 2: **Mean post-training accuracy** for fast-learning and natural MothNet moths, as well as for Nearest-neighbors, SVM, and CNN, vs number of training samples (log scale). In the few-training-samples regime (1 to 20 per class), MothNet substantially outperformed standard ML methods. $N = 11$ per data point.

## 4 DISCUSSION

In order to learn new odors, the moth olfactory network uses just a few core tools: A noisy pre-amp network with competitive inhibition; Hebbian plasticity controlled by a high-dimensional sparse layer; and generalized (global) stimulation during training. Our key finding is that the simplest of BNNs (an insect brain), built with this biological toolkit, can succeed at a general learning task, and in fact can out-perform standard ML methods.

The biological tools analyzed here are well-suited to being combined and stacked into larger, deeper neural nets, just as convolutional kernels, maxpool, etc, are combined to build current DNNs. The success of live BNNs at a wide range of tasks argues for the potential of NNs built with a biological toolkit to succeed at ML tasks.

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
