# OpenReview forum: "A moth brain learns to read MNIST"
_ICLR.cc/2018/Workshop — Accept_

### Official Review · AnonReviewer3 · 2018-03-05
**Interesting model for few-shot learning but partly unclear; Comparison to other few-shot learning models missing.**

**Rating:** 6
**Confidence:** 4

**Review:**

The authors present a simple neural network architecture inspired by the moth's olfactory system. They show that, when combined with Hebbian type learning rules in a supervised setup it can perform few-shot learning.
In particular, they show that it can learn to classify MNIST digits with an accuracy of 75 % to 85 % on test digits after training on 1 to 20 samples per class. Performance is compared to standard classifiers (SVMs, nearest neighbor classifiers, and convolutional neural networks)

In general, the paper is interesting, showing that a quite simple neural network can achieve reasonable performance in few-shot learning. It is hard to judge how well the network performs, as it is only compared to methods that are not optimized for few-shot learning. A comparison with few-shot learning models or meta-learning methods seems necessary.
Also, while the model is described in quite some detail, the description is unclear at several places, including one detail that seems important to judge the achievements of the model.
The writing could be improved regarding mathematical notation and clarity.
A more detailed analysis of the model is missing. It would be good to study what components of the model are essential in order to achieve the reported results.

Some details:
p.3:
- What is the reference set V? This seems important. Does it consist of examples that have been used for training or is it an additional set. If the latter, how large is it? If it is the latter, the conclusions are questionable, as one actually needs more examples from the dataset than stated.
p.2:
- I would not call (1) "integrate-and-fire dynamics" as there is no firing. It is rather a (stochastic) leaky integrator dynamics.
- Indicate the variable over which the sum runs in (1).
- Is the range of the sigmoid S non-negative or are negative outputs possible?
- In (1), weights have a single index, in (2) and (3) they have two indices.
- Training data: The transformation of the training data is unclear. What does "killed negative values" mean?

---

### Official Review · AnonReviewer2 · 2018-03-09
**Review: A moth brain learns to read MNIST**

**Rating:** 5
**Confidence:** 4

**Review:**

In this paper, the authors present an application of a recently developed firing rate model of the Moth Olfactory Network (MothNet) to MNIST. The authors introduce the MothNet model, plasticity, training data, and their classifier. They then go on to show that the MothNet model can classify MNIST digits well in the low training samples regime.

Overall, the results are interesting but incomplete. Why does the accuracy testing for the MothNet stop at 20 samples? If the other ML methods (e.g. CNN) outperform this algorithm, it is important to show this directly and explain the results. The fast learning performance of the two MothNet versions tested is interesting, but the fact that the learning stops after essentially one sample is a problem for the algorithm. A more general understanding of the model and the results would perhaps lead to a more clear workshop proposal.

---

### Decision · Program_Chairs · 2018-03-20
**ICLR 2018 Workshop Acceptance Decision**

**Decision:**

Accept

**Comment:**

Congratulations, your paper was accepted to the ICLR workshop.